# Sunburns and Sun Protection Behaviors among Male Hispanic Outdoor Day Laborers

**DOI:** 10.3390/ijerph19052524

**Published:** 2022-02-22

**Authors:** Zhaomeng Niu, Mary Riley, Jerod L. Stapleton, Michele Ochsner, Germania Hernandez, Louis Kimmel, Daniel P. Giovenco, Shawna V. Hudson, Denalee O’Malley, Carolina Lozada, Marién Casillas Pabellón, Carolyn J. Heckman, Elliot J. Coups

**Affiliations:** 1Rutgers Cancer Institute of New Jersey, New Brunswick, NJ 08901, USA; zhaomeng.niu@rutgers.edu (Z.N.); hudsonsh@rwjms.rutgers.edu (S.V.H.); omalledm@rwjms.rutgers.edu (D.O.); lozadaci@cinj.rutgers.edu (C.L.); 2Medtronic, Boulder, CO 80301, USA; rileymk12@gmail.com; 3College of Public Health, University of Kentucky, Lexington, KY 40506, USA; jerod.stapleton@uky.edu; 4Department of Labor Studies and Employment Relations, School of Management and Labor Relations, Rutgers, The State University of New Jersey, New Brunswick, NJ 08901, USA; mochsner@work.rutgers.edu; 5New Labor, New Brunswick, NJ 08901, USA; germaniah@newlabor.org (G.H.); lkimmel@newlabor.org (L.K.); 6Department of Sociomedical Sciences, Mailman School of Public Health, Columbia University, New York, NY 10032, USA; dg2984@cumc.columbia.edu; 7Department of Family Medicine and Community Health, Rutgers Robert Wood Johnson Medical School, Rutgers, The State University of New Jersey, New Brunswick, NJ 08901, USA; 8Interfaith Worker Justice, Chicago, IL 60660, USA; marien@pasoaction.org; 9Department of Medicine, Rutgers Robert Wood Johnson Medical School, Rutgers, The State University of New Jersey, New Brunswick, NJ 08901, USA

**Keywords:** UV exposure, sun protection, sunburn, Hispanic, Latino, outdoor day laborers, skin cancer prevention, heat illness

## Abstract

Individuals who work outside are at increased risk for skin cancer due to excessive exposure to ultraviolet (UV) radiation. Little is known about UV exposures and sun safety practices of outdoor day laborers, who are disproportionately Hispanic. This study identified the correlates of sunburn and sun protection behaviors in a sample of male, Hispanic day laborers (*n* = 175). More than half of the participants (54.9%) experienced one or more sunburns when working during the past summer, and 62.9% reported having one or more symptoms of heat illness. The frequency of engaging in sun protection behaviors was suboptimal, including sunglasses use (*M* = 2.68, *SD* = 1.71), staying in the shade (*M* = 2.30, *SD* = 0.94), wearing sunscreen (*M* = 2.10, *SD* = 1.39), and wearing a wide-brimmed hat (*M* = 1.75, *SD* = 1.32), based on a 5-point scale (1 = never; 5 = always). Lower education level, higher levels of skin sensitivity to the sun, any symptom of heat illness, fewer barriers to wearing a wide-brimmed hat, and not wearing a wide-brimmed hat were associated with a greater number of sunburns. Factors associated with each sun protection behavior varied. Implications and directions for future research are discussed.

## 1. Introduction

Melanoma incidence has risen in the past decade [1]. Although the Hispanic population has lower incidence rates of both melanoma and keratinocyte cancers compared to non-Hispanic whites (NHW), the mortality rate of Hispanics is higher [1]. In general, compared to NHW, Hispanics diagnosed with melanoma are younger (median age: Hispanics—49 years; NHW—58 years), have advanced stages of melanoma with thicker tumors during diagnosis (>1.5 mm), and have worse 5-year melanoma-specific survival rates (for men, 5-year survival of 76.6% and 87.0%, respectively; for women, 88.3% and 92.3%) [2,3,4,5]. Overexposure to the sun is the leading cause of skin cancer. Over 86% of melanoma and 90% of keratinocyte skin cancers are attributed to exposure to ultraviolet (UV) radiation from the sun [6]. Individuals who work outside are at increased risk for skin cancer due to regular, excessive exposure to ultraviolet (UV) radiation from the sun [7,8]. Hispanic individuals in the United States are overrepresented in numerous outdoor occupations that have high UV exposure, such as landscaping, construction, and farming [9]. There are more than 100,000 day laborers in the United States, the vast majority of whom are Hispanic men [10]. Several prior studies have focused on work conditions, job-related hazards, and limited protections afforded to the Hispanic day laborer workforce [10,11,12,13]. The U.S. Surgeon General’s Call To Action to Prevent Skin Cancer highlighted the need for research to understand sun protection behaviors of outdoor workers from varying cultures and backgrounds and recommended educational outreach for outdoor workers to increase awareness about the risks of occupational sun exposure [14].

Yet, information regarding sunburns and sun protection behaviors among Hispanic outdoor day laborers, especially in the northeast and working in primarily urban areas, is lacking. Prior research also suggests that sun protection behaviors (e.g., sunscreen use, sun protective clothing, staying in the shade, etc.) among Hispanics are insufficient [15,16,17]. A recent study of Hispanic migrant farmworkers (*n* = 157) in North Carolina reported low engagement in sun safety practices, including sunscreen use (9.2%), sunglasses use (11.3%), and wearing wide-brimmed hats (27.5%) [18]. In this study, consistently wearing long-sleeved shirts was the most common sun protective behavior reported (85.7%). This study also found that knowledge about sun protection and skin cancer was very low in the migrant farmworkers population. These results are largely consistent with those from a prior study of 326 Hispanic farmworkers in California [19]. Additionally, in a sample of 149 Hispanic outdoor workers drawn from a population-based survey of individuals residing in five southern and western states, we found that only 19.0% of participants reported often or always using sunscreen when working outside in the sun [20]. A higher frequency of sunscreen use was found among women, those with a higher level of education, and individuals who resided at a higher latitude. Further, Boyas and Nahar [21] studied sun safety among a regional sample of Latino day laborers (*n* = 137) from Mississippi and Illinois and found that Latino day laborers had marginal levels of skin cancer knowledge and engaged minimally in sun protective behaviors. In the current study, we focused on a specific subgroup of Hispanic outdoor day laborers, who typically seek work in informal, open-air urban venues such as parking lots and street corners.

The aim of the current study was to identify and describe correlates of sunburns and sun protection behaviors among Hispanic outdoor day laborers. The Preventive Health Model (PHM) was used as a guiding theoretical framework to examine potential correlates of sun protection and sunburns in the current study [22]. This model suggests that individuals’ engagement in preventive behaviors is influenced by multiple determinants, including background (e.g., demographic variables), affective, cognitive, and social factors [22]. This model has been utilized to predict skin cancer-related preventive behaviors (e.g., sun protection) in previous studies [23,24,25] and has also been used to investigate psychological correlates of sun protection behaviors among Hispanics in the US [15,16,26,27]. This study can inform future policies, outreach interventions, and research to mitigate the risks of sun exposure in this population.

## 2. Materials and Methods

### 2.1. Participants

Individuals were eligible to participate if they were male, self-identified as Hispanic or Latino, and reported working outside in the past 3 months or during the previous summer. A total of 175 individuals completed the survey.

### 2.2. Data Collection

Study participants were recruited by staff members of New Labor, a non-profit, community-based worker organization in the city of New Brunswick, NJ, USA. Participants were approached at local outdoor locations where day laborers commonly congregate to obtain work. Individuals who had previously participated in worker trainings or educational sessions offered by New Labor were also invited to participate in the study. Participants completed a 15–20-min pencil and paper or interviewer-administered survey and received $15 for their participation. All participants completed the survey in Spanish (an English version was also available). Informed consent was obtained from all participants prior to completing the survey. This study was reviewed and approved by the Rutgers Health Sciences Institutional Review Board.

### 2.3. Measures

Survey items that were not already available in Spanish were professionally translated and then refined for plain language adaptation by bilingual research staff members. This model suggests that individual’s engagement in preventive behaviors is influenced by multiple determinants including background (e.g., demographic variables), affective, cognitive, and social factors [22].

Sociodemographic factors. Participants reported their age, years of schooling, Hispanic heritage, preferred language, and the number of years they had lived in the United States.

Skin sensitivity. A single item, commonly used in research including with Hispanics, assessed skin sensitivity to the sun [16]. Specifically, participants indicated how their skin would react if they stayed outdoors in the midday summer sun for one hour without sun protection (using a 5-point response scale from 5 = severe sunburn to 1 = nothing would happen to my skin).

Outdoor work. Participants reported the number of days that they worked outdoors in a typical week in the past summer. The average UV index in New Jersey the summer before data collection was approximately 9, which is considered “very high”. They also indicated the types of work that they performed (e.g., construction, moving/hauling, and gardening/landscaping).

Knowledge of sun protection and skin cancer. Seven items adapted from prior research [28] assessed their knowledge of sun protection and skin cancer (e.g., “Most skin cancers are caused by sun exposure”). For each item, the response options were true, false, and don’t know. A total knowledge score (ranging from 0 to 7) was calculated for each participant by summing the number of correct answers (with a don’t know response being coded as incorrect).

Sun protection and skin cancer beliefs. Using a 5-point response scale (from 1 = strongly disagree to 5 = strongly agree), participants completed single-item measures of perceived natural skin protection, appearance benefits of a tan, sun protection importance, perceived photoaging risk, perceived skin cancer risk, and skin cancer fatalism [26,29,30].

Sun protection barriers. Participants completed a checklist of potential barriers to using sunscreen (e.g., “It is unpleasant” and “It interferes with my work activities”), wearing long pants and a long-sleeved shirt (e.g., “It is too hot” and “Other workers don’t wear them”), and wearing a wide-brimmed hat (e.g., “It is embarrassing” and “Bosses say it isn’t necessary”) when they work outside on a warm, sunny day. Ten barriers were listed for sunscreen and eight for the other behaviors. The barriers for each behavior were drawn from prior research [26,31] and, also, from the results of several focus groups that we conducted with male Hispanic outdoor day laborers prior to conducting this survey (data not shown).

Sun protection behaviors. The frequency (using a 5-point response scale from 1 = never to 5 = always) of sun protection behaviors (e.g., sunscreen use, worked in the shade, wore a long-sleeved shirt, wore long pants, wore a wide-brimmed hat, wore a baseball cap, and wore sunglasses) was assessed based on self-reported practices on warm, sunny days during the previous summer [32]. Individuals who reported using sunscreen were asked whether they reapplied it when they were outside for a long time.

Heat illness symptoms. The number of times last summer participants felt weak, dizzy, nauseous, or had a headache from working outside in the sun was assessed.

Sunburns. Participants reported the number of times they had a sunburn last summer from working outside in the sun.

### 2.4. Statistical Analyses

Descriptive analyses examined the characteristics of the sample, sun protection and skin cancer knowledge, sun protection and skin cancer beliefs, sun risk and protection behaviors, and sun protection barriers. Occupational variables were examined using logistic regression, and construction work was the only predictor that was significantly associated with sunburns; therefore, construction was entered into the subsequent adjusted logistic regression model. Additionally, we conducted a logistic regression to examine the correlates of having had one or more sunburns during the past summer and a set of multivariable linear regressions to test the potential correlates of different sun protection behaviors. Participants’ age, years of schooling, Hispanic heritage, preferred language, number of years they had lived in the United States, and their skin sensitivity to the sun were controlled in the regression analyses. A value of *p* < 0.05 was used to determine statistical significance.

## 3. Results

The characteristics of the study sample (N = 175) are shown in Table 1. The sample had an average age of 35 years, with almost half of participants identifying with Mexican heritage. Participants reported the average amount of time living in the United States as 12.3 years. The most frequent work activities among the sample were construction, landscaping/gardening, and painting. Participants worked 5.2 days a week outside on average in the past summer. Over 80% of respondents reported that their skin reacts with at least a mild sunburn when they work outside in the midday summer sun for an hour. More than half of participants reported at least one sunburn (54.8%) or episode of heat illness during the previous summer (62.9%).

### 3.1. Sun Protection and Skin Cancer Knowledge

There were clear deficits in sun protection and skin cancer knowledge in the sample. The majority of respondents demonstrated basic knowledge about sun-related cancer risks—that sunburns increase skin cancer risk (true, 88.6%) and only people with light skin need to protect themselves from the sun (false, 85.1%). Regarding protective behaviors, only 2.3% of respondents accurately assessed the question about using sunscreen with a sun protection factor (SPF) number of at least 10 (false). Details of all seven items of knowledge are shown in Table 2.

### 3.2. Sun Protection and Skin Cancer Beliefs

More than two-thirds of the participants indicated that they neither believe that their natural skin color protects them from the sun (*n* = 137, 78.3%) nor that they look better with a tan (*n* = 119, 68%). Most of the participants agreed that there were things they could do to lower their chances of getting skin cancer (*n* = 143, 81.7%). More than 80 percent of the sample endorsed: (1) the importance of protecting their skin from the sun while working outdoors (*n* = 160, 91.4%), (2) that their skin would be likely to be damaged (*n* = 163, 93.1%), and (3) their chances of getting skin cancer in their lifetime would be high (*n* = 146, 83.4%) if they did not protect their skin from the sun while working outdoors. The details are shown in Table 3.

### 3.3. Sun Protection Barriers

Among the barriers that participants reported to influence their practice of sun protection, feeling uncomfortable and “too hot” were the most commonly listed barriers to wearing a wide brimmed hat (40.6% and 24.0%), long pants, and/or a long-sleeved shirt (25.1% and 33.1%) on warm, sunny days. Forgetting to apply sunscreen (28.0%) and a lack of knowledge of effective sunscreen products to use (22.9%) were highly endorsed barriers of using sunscreen when working outside, followed by perceiving that applying sunscreen was unpleasant (15.4%) and expensive (12.0%). Detailed results are displayed in Table 4.

### 3.4. Sun Protection Behaviors

Among individuals who reported using sunscreen (*n* = 82, 46.9% of the sample), less than half (42.7%) indicated that they reapplied it when they are outside for a long time (see Table 5). In general, participants did not engage in sufficient sun protection behaviors, especially wearing sunglasses (*M* = 2.68, *SD* = 1.71), staying in the shade (*M* = 2.30, *SD* = 0.94), wearing sunscreen (*M* = 2.10, *SD* = 1.39), or wearing a wide-brimmed hat (*M* =1.75, *SD* = 1.32). The average frequency of engagement in sun protection behaviors (i.e., staying in the shade, wearing a long-sleeved shirt, long pants, a baseball cap, a wide-brimmed hat, sunglasses, or sunscreen) reported by the participants was 2.88 (*SD* = 0.64). Approximately two-thirds of participants stated that they never or rarely wore sunscreen (66%) or stayed in the shade (61%) when working outside in the past summer.

### 3.5. Correlates of Sunburn

The results of the logistic regression indicated that non-Mexican heritage, years of education, skin sensitivity to sun, frequency of heat illness last summer, barriers to wearing a wide-brimmed hat, wearing a wide-brimmed hat, and working in construction were significant correlates of having one or more sunburns last summer. Compared to participants with Mexican heritage, those who indicated “other” Hispanic heritages (e.g., Puerto Rican and Dominican) were more likely to report sunburn in the previous summer (adjusted odds ratio (AOR) = 3.59, 95% CI (confidence interval): 1.05–12.34). Respondents who had higher education levels reported fewer sunburns (AOR = 0.89, 95% CI: 0.786–0.998). Respondents who anticipated severe sunburn (AOR = 63.26, 95% CI: 6.93–577.38), moderate sunburn (AOR = 17.34, 95% CI: 2.35–128.18), mild sunburn (AOR = 24.36, 95% CI: 3.13–189.57), and darker skin without sunburn (AOR = 590.96, 95% CI: 22.68–15398.01) were more likely to experience sunburns than those who believed nothing would happen to their skin if they remained unprotected during a mid-summer sunny day. Participants reporting one heat illness (AOR = 15.74, 95% CI: 3.87–64.05) or more than one (AOR = 20.64, 95% CI: 6.04–70.52) in the past summer were more like to have experienced a sunburn than those who had no heat illness. Interestingly, individuals who reported more barriers to wearing a wide-brimmed hat reported being less likely to have sunburns (AOR = 0.47, 95% CI: 0.23–0.95). However, respondents who did not wear wide-brimmed hats had a greater likelihood of at least one sunburn compared those reporting hat use (AOR = 0.66, 95% CI: 0.44–0.99). Lastly, participants who reported working in construction last summer were more likely to have sunburns (AOR = 6.37, 95% CI: 2.31–17.51) than other types of workers. The details are shown in Table 6.

### 3.6. Correlates of Sun Protection Behaviors

A separate multivariable regression was conducted to explore the potential correlates of each sun protection behavior among the participants (Table 7). Individuals who worked more outside in the past summer stayed in the shade less (B = −0.24, *p* < 0.001). Participants who had greater barriers to wearing protective clothing (B = −0.25, *p* = 0.001) and thought they looked better with a tan (B = −0.10, *p* = 0.03) were less likely to wear long pants. Individuals who had more years of education (B = −0.07, *p* = 0.018) and had greater barriers to wearing protective clothing (B = −0.68, *p* < 0.001) were less likely to wear a long-sleeved shirt. Participants who reported having symptoms of heat illness, including feeling weak, dizzy, nauseous, or had a headache from working outside in the sun last summer (B = 0.25, *p* = 0.028); having few barriers to wearing a hat (B = −0.22, *p* = 0.018); and thinking it is important to protect their skin from the sun while working outdoors (B = 0.29, *p* = 0.027) were more likely to report wearing wide-brimmed hats. Respondents who thought that the natural color of their skin protected them from the sun were more likely to wear sunglasses (B = 0.19, *p* = 0.042). Those who thought their skin looked better with a tan were more likely to wear sunscreen (B = 0.14, *p* = 0.046).

## 4. Discussion

The melanoma mortality rate of Hispanics is higher than whites [1]. Individuals who work outside experience occupational sun exposure are at increased risk for skin cancer [7,8], and they do not sufficiently engage in sun protection behaviors [15,16,17]. The current study provided important data regarding Hispanic outdoor workers’ sun protection behaviors, sunburns, and skin cancer-related perceptions.

The current study found that more than half of male Hispanic outdoor day laborers reported having at least one sunburn or at least one episode of heat illness during the past summer. Having sunburns increases the risk of getting melanoma [33,34]. Individuals who identified with non-Mexican Hispanic heritages other than Honduran (e.g., Dominican and Puerto Rican) were more likely to get sunburns compared to those identifying with Mexican heritage. This finding confirms the importance of attending to the heterogeneity among the Hispanic population. Participants with higher education levels also reported fewer sunburns, which suggested the important role of education in preventing sunburns among male Hispanic outdoor laborers [35]. Participants’ skin reaction in summer sun for one hour without protection was significantly associated with their history of sunburn. Those individuals whose skin would have any sunburn and turn darker without sunburn were more likely to report having sunburns. Another physiological factor that was associated with having a sunburn was heat illnesses. Individuals who had felt weak, dizzy, nauseous or had a headache from working outside in the sun were more likely to have sunburns. These findings indicated that male Hispanic outdoor day laborers who have sensitive skin to sun exposure or who have a history of heat illness are at risk for excessive UV exposure and sunburns. Individuals who reported wearing a wide-brimmed hat were less likely to have a sunburn, which indicated that wearing a wide-brimmed hat may be particularly effective at preventing sunburns among male Hispanic outdoor day laborers. Working in construction was also associated with sunburns, which suggested that male Hispanic outdoor day laborers working in construction industry are particularly in need of sun protection. Future skin cancer education programs or health interventions are needed for this group to reduce occupational UV exposure [14]. One seemingly paradoxical finding indicated that individuals who reported more barriers to wearing wide-brimmed hat were less likely to have sunburns. One possible explanation could be that individuals who had greater barriers to wearing wide-brimmed hats tried to engage more in other types of sun protection behaviors.

The results also suggested that male Hispanic outdoor day laborers do not engage in sufficient sun protection behaviors. This finding is consistent with a previous study indicating that Hispanics in New York and Florida do not engage in sufficient sun protection activities [36]. The average frequency of engagement in sun protection behaviors overall (i.e., staying in the shade, wearing a long-sleeved shirt, long pants, a baseball cap, a wide-brimmed hat, sunglasses, or sunscreen) reported by male Hispanic outdoor workers in the present study ranged from rarely to sometimes. The differences in frequencies among different sun protection behaviors are consistent with previous observations among Hispanic adults and Hispanic US adults [26,27]. Wearing a wide-brimmed hat was associated with fewer sunburns, suggesting appropriate sun protection behaviors can reduce sunburns among male Hispanic outdoor day laborers. Participants’ tanning beliefs (skin looks better with a tan) were only associated with wearing long pants and wearing sunscreen. This finding may suggest that, in the Hispanic population, tanning beliefs may work differently in sun protection behaviors compared to the white population. Previous studies have suggested that Hispanics or Asians seem not to have favorable attitudes towards tanned skin [37,38]. Future studies could develop some items to measure more tanning attitudes among the Hispanic population. Our findings indicate that investigations of skin cancer interventions on sun protection behaviors among male Hispanic outdoor day laborers are needed. Future research targeting behavioral change regarding Hispanic outdoor day laborers employing effective messages and interventions is warranted.

This is the first study, to our knowledge, to examine both sunburns and sun protection behaviors among male Hispanic outdoor day laborers in the Northeast US. Our study is also unique, because we combined the knowledge about sun protection with using sun protection in practice. It is important to investigate different types of occupations of outdoor workers [39,40]. The strengths of this study included detailed information on Hispanic heritage, occupation type, and inclusion of occupation-related sun protection variables such as sun protection barriers. The findings related to factors that were significantly associated with Hispanic outdoor workers’ sun protection behaviors and sunburns helped identify components to target in future skin cancer interventions among male Hispanic outdoor day laborers. The current study also had several limitations. Given the cross-sectional nature of the survey, we could not make inferences about causality. The inclusion of only male Hispanic outdoor day laborers from New Jersey may limit the findings’ generalizability. The data were based on self-reported information, which was subject to bias. Future studies could aim to measure sun exposure using objective measures among Hispanic outdoor day laborers. The nature of outdoor occupations results in different UV exposures across work types and settings (e.g., construction vs. gardening). Therefore, future studies should assess the differences in UV exposure between various outdoor occupations.

## 5. Conclusions

This study examined both sunburns and sun protection behaviors among male Hispanic outdoor day laborers in the Northeast US. The findings of this study contribute to the scientific knowledge of sunburns and sun protection behaviors among an understudied area of research. The study suggested that participants did not engage in sufficient sun protection behaviors. Moreover, a lower education level, higher levels of skin sensitivity to the sun, any symptom of heat illness, fewer barriers to wearing a wide-brimmed hat, and not wearing a wide-brimmed hat were associated with a greater number of sunburns. The factors associated with specific sun protection behaviors varied by each behavior. The study results directly inform the need for and potential development of interventions to promote sun protection behaviors among male Hispanic outdoor day laborers in future research.

## Figures and Tables

**Table 1 ijerph-19-02524-t001:** Characteristics of the study sample (N = 175).

	Mean (SD)	Sample %
Age, years	35.1 (9.1)	
18–29		29.1
30–39		38.9
≥40		32.0
Country of Origin		
Mexico		48.0
Honduras		26.9
Dominican Republic		6.4
Puerto Rico		4.1
Guatemala		4.1
El Salvador		3.5
Ecuador		2.9
Peru		2.3
Nicaragua		1.2
Costa Rica		0.6
Years of Education	9.6 (4.1)	
≤5		12.0
6–8		26.9
9–11		24.6
≥12		36.6
Years Living in the U.S.	12.3 (7.9)	
≤5		18.3
6–10		30.9
11–20		38.3
>20		12.6
Days/Week Worked Outside Last Summer	5.2 (1.2)	
1		0.0
2		2.3
3		9.7
4		8.6
5		37.7
6		30.3
7		11.4
Type of Work Performed in the Past Summer ^a^		
Construction		59.4
Gardening/Landscaping		58.3
Painting		26.9
Moving/hauling		14.9
Drywall installation		12.6
Carpentry		10.9
Masonry		9.1
Car washing		8.0
Plumbing		7.4
Roofing		6.7
Farm work		5.1
Warehouse worker		4.6
Restaurant worker		3.4
Electrician		2.3
Other work		22.9
Skin Sensitivity to the Sun ^b^		
Severe sunburn		15.4
Moderate sunburn		35.4
Mild sunburn		29.7
Turn darker without sunburn		6.3
Nothing would happen to my skin		13.1
Number of Times Had Any Symptoms of Heat Illness in the Past Summer ^c^		
0		37.1
1		16.6
≥2		46.3
Number of Sunburns in the Past Summer		
0		45.1
1		19.4
≥2		35.4

^a^ Participants could endorse more than one type of work performed in the past summer. ^b^ Participants reported how their skin would react if they stayed outdoors in the midday summer sun for one hour without sun protection. ^c^ Participants reported the number of times they felt weak, dizzy, nauseous, or had a headache from working outside in the sun.

**Table 2 ijerph-19-02524-t002:** Knowledge of sun protection and skin cancer.

Items	% Correct
Getting sunburns increases the risk of skin cancer ^a^	88.6
Only people with light skin need to protect themselves from the sun ^b^	85.1
People with dark skin cannot get skin cancer ^b^	76.0
Most skin cancers are caused by sun exposure ^a^	68.0
Most wrinkles in the skin are caused by sun exposure ^a^	48.6
Light colored clothing provides more protection against the sun than dark colored clothing ^b^	23.4
It is recommended that people use sunscreen with a sun protection factor (SPF) number of at least 10 ^b^	2.3

^a^ The statement is true. ^b^ The statement is false.

**Table 3 ijerph-19-02524-t003:** Beliefs regarding sun protection and skin cancer.

Items	Somewhat/ Strongly Disagree %	Neither Agree Nor Disagree %	Somewhat/ Strongly Agree %	Mean (SD)
The natural color of my skin protects me from the sun	78.3	4.6	17.1	1.81 (1.50)
My skin looks better with a tan	68.0	3.4	28.6	2.19 (1.69)
It is important for me to protect my skin from the sun while working outdoors	7.4	1.1	91.4	4.65 (1.04)
If I don’t protect myself from the sun while working outdoors, my skin will be very likely to be damaged	5.1	1.7	93.1	4.71 (0.91)
If I don’t protect my skin from the sun while working outdoors, I feel that my chances of getting skin cancer in my lifetime are high	9.7	6.9	83.4	4.43 (1.18)
There’s not much you can do to lower your chances of getting skin cancer	81.7	6.9	11.4	1.58 (1.15)

**Table 4 ijerph-19-02524-t004:** Barriers to engaging in sun protection behaviors when working outside on warm, sunny days.

Items	Sample %
**Barriers to Using Sunscreen**	
I forget to apply it	28
I don’t know what kind to use	22.9
It is unpleasant	15.4
It is expensive	12
None of the above	38.3
**Barriers to Wearing Long Pants and a Long-Sleeved Shirt**	
It is too hot	33.1
It is uncomfortable	25.1
It interferes with my work activities	6.9
I have to wear a work uniform	4
None of the above	50.9
**Barriers to Wearing a Wide-Brimmed Hat**	
It is uncomfortable	40.6
It is too hot	24
I don’t like it	13.7
It interferes with my work activities	10.9
I have to wear a work uniform	5.1
None of the above	38.9

Participants could endorse more than one barrier for each behavior.

**Table 5 ijerph-19-02524-t005:** Sun protection behaviors when working outside on warm, sunny days last summer.

Items	Never/Rarely %	Sometimes %	Often/Always %	Mean (SD)
Wear sunscreen	66.0	17.0	17.0	2.10 (1.39)
Stay in the shade	61.0	29.0	10.0	2.30 (0.94)
Wear a long-sleeved shirt	36.0	20.0	44.0	3.26 (1.55)
Wear long pants	6.9	4.5	88.6	4.65 (0.93)
Wear a wide-brimmed hat	78.9	8.0	13.1	1.75 (1.32)
Wear a baseball cap	36.0	9.7	54.3	3.39 (1.78)
Wear sunglasses	54.3	12.0	33.7	2.68 (1.71)

**Table 6 ijerph-19-02524-t006:** Logistic regression of the correlates of sunburns.

Variable	B	SE	*p*	Exp(B)	95% CI for EXP(B)	
					Lower	Upper
Age	−0.040	0.032	0.212	0.961	0.903	1.023
Hispanic heritage			0.074			
Mexican			Ref			
**Other**	**1.279**	**0.629**	**0.042**	**3.594**	**1.046**	**12.341**
Honduran	−0.346	0.598	0.563	0.708	0.219	2.284
**Years of education**	**−0.121**	**0.061**	**0.047**	**0.886**	**0.786**	**0.998**
Years lived in the United States	0.000	0.037	0.999	1.000	0.930	1.075
Language Acculturation			0.268			
Only Spanish			Ref			
Mostly Spanish	−0.914	0.571	0.110	0.401	0.131	1.229
Spanish/English about the same or mostly English	−0.673	0.691	0.330	0.510	0.132	1.978
**Skin Sensitivity to the Sun**			0.001			
Nothing would happen to my skin			Ref			
**Severe sunburn**	**4.147**	**1.128**	**0.000**	**63.260**	**6.931**	**577.380**
**Moderate sunburn**	**2.853**	**1.021**	**0.005**	**17.337**	**2.345**	**128.184**
**Mild sunburn**	**3.193**	**1.047**	**0.002**	**24.357**	**3.130**	**189.565**
**Turn darker without sunburn**	**6.382**	**1.663**	**0.000**	**590.958**	**22.680**	**15,398.012**
Days per week worked outside this past summer	0.203	0.202	0.314	1.226	0.825	1.820
**Heat illness**			0.000			
0 times			Ref			
**1 time**	**2.756**	**0.716**	**0.000**	**15.736**	**3.866**	**64.049**
**2 or more times**	**3.027**	**0.627**	**0.000**	**20.644**	**6.044**	**70.515**
Knowledge	0.224	0.248	0.367	1.251	0.769	2.034
Barriers to using sunscreen	0.136	0.242	0.575	1.146	0.712	1.842
Barriers to wearing clothes	0.135	0.301	0.654	1.145	0.634	2.064
**Barriers to wearing a wide-brimmed hat**	**−0.762**	**0.364**	**0.036**	**0.467**	**0.229**	**0.952**
The natural color of my skin protects me from the sun	−0.337	0.175	0.054	0.714	0.506	1.006
My skin looks better with a tan	0.064	0.155	0.681	1.066	0.786	1.445
It is important for me to protect my skin from the sun while working outdoors	−0.017	0.334	0.960	0.983	0.511	1.892
If I don’t protect my skin from the sun while working outdoors, my skin will be very likely to be damaged	0.014	0.417	0.973	1.014	0.448	2.297
If I don’t protect my skin from the sun while working outdoors, I feel that my chances of getting skin cancer in my lifetime are high	−0.012	0.310	0.969	0.988	0.539	1.813
There’s not much you can do to lower your chances of getting skin cancer	0.033	0.256	0.896	1.034	0.627	1.706
Sun protection						
Work in the shade	0.314	0.275	0.254	1.369	0.798	2.348
Wear a long-sleeved shirt	0.055	0.200	0.783	1.057	0.714	1.563
Wear long pants	−0.171	0.263	0.516	0.843	0.503	1.412
Wear a baseball cap	−0.067	0.135	0.621	0.935	0.717	1.219
**Wear a wide-brimmed hat**	**−0.417**	**0.209**	**0.046**	**0.659**	**0.437**	**0.993**
Wear sunglasses	−0.001	0.138	0.997	0.999	0.762	1.310
Wear sunscreen	0.088	0.178	0.620	1.092	0.770	1.549
**Construction**	**1.851**	**0.516**	**0.000**	**6.365**	**2.314**	**17.506**

Boldface indicates significant association. CI = confidence interval, B = beta logistic regression coefficient, SE = standard error, and Exp(B) = exponentiation of the B coefficient, which is an odds ratio.

**Table 7 ijerph-19-02524-t007:** Significant correlates of sun protection behaviors among Hispanic outdoor day laborers.

Sun Protection Item ^a^	Significant Predictor	B	SE	Beta	t	*p*
Work in the shade	Days per week worked outside this past summer	−0.244	0.060	−0.308	−4.050	0.000
Wear a long-sleeved shirt ^b^	Years of education	−0.065	0.027	−0.171	−2.389	0.018
	Barriers to wearing long pants and a long-sleeved shirt	−0.682	0.106	−0.437	−6.408	0.000
Wear long pants ^b^	Barriers to wearing long pants and a long-sleeved shirt	−0.250	0.070	−0.266	−3.550	0.001
	My skin looks better with a tan	−0.097	0.045	−0.175	−2.155	0.033
Wear a baseball cap	None					
Wear a wide-brimmed hat ^c^	Heat illness	0.249	0.113	0.173	2.216	0.028
	Barriers to wearing a wide-brimmed hat	−0.219	0.091	−0.195	−2.398	0.018
	It is important for me to protect my skin from the sun while working outdoors	0.293	0.131	0.231	2.226	0.027
Wear sunglasses	The natural color of my skin protects me from the sun	0.191	0.093	0.168	2.048	0.042
Wear sunscreen ^d^	My skin looks better with a tan	0.141	0.070	0.172	2.009	0.046

^a^ Variables in all regression models included: age, heritage, year of education, years lived in the US, language acculturation, skin sensitivity to the sun, days per week worked outside this past summer, heat illness, knowledge, and six beliefs regarding sun protection and skin cancer items. ^b^ Barriers to wearing long pants and a long-sleeved shirt were included in the regression models of wearing sunscreen and wearing long pants. ^c^ Barriers to wearing a wide-brimmed hat were only included in the regression model of wearing a wide-brimmed hat. ^d^ Barriers to using sunscreen were only included in the regression model of wearing sunscreen. B = beta unstandardized regression coefficient, SE = standard error, and Beta = standardized regression coefficient.

## Data Availability

The de-identified datasets analyzed during the current study are available from the corresponding author upon reasonable request.

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
