# Peer review of "Sunburns and Sun Protection Behaviors among Male Hispanic Outdoor Day Laborers"

_ijerph, 2022, doi:10.3390/ijerph19052524_

Round 1

Reviewer 1 Report

The article "Sunburns and Sun Protection Behaviors among Male Hispanic Outdoor Day Laborers" by Niu et al. consists of 14 pages. The abstract is well written and well summarizes the research. Research about outdoor day laborers and their patterns of sun-exposure behavior is of great importance but it is nothing new. Nevertheless, the interesting point of the study is combining the knowledge about sun protection with using sun protection in practice.

The study examines different types of occupations, but do not consider, that different types of occupations mean different poses towards the sun. It also does not give the information about erythemal doses, that workers gained during their exposure. At least the information about maximum UV Index in the summer in the place of working should be included in the text. The paper should be reconsidered after the minor revision.

Detailed comments:

2.1, page 3 - what was the period of the study? Which years were examined? What was the maximum UV Index?

Line 120, page 3: Why the authors did not use the Fitzpatrick scale?

Line 241, page 8: Authors explained the shortcut AOR, but not others (CI, B, S.E., Exp(B)). These shortcuts may be obvious to the authors, but not to the readers, who are not familiar with this type of regression.

Line 338, page 12: It is not novel to investigate different types of occupations. See the references (and also the list of references in these papers):

Schmalwieser, A.W.; Casale, G.R.; Colosimo, A.; Schmalwieser, S.S.; Siani, A.M. Review on Occupational Personal Solar UV Exposure Measurements. Atmosphere 202112, 142. https://doi.org/10.3390/atmos12020142 

Siani, A.M., Casale, G.R., Sisto, R., Colosimo, A., Lang, C.A. and Kimlin, M.G. (2011), Occupational Exposures to Solar Ultraviolet Radiation of Vineyard Workers in Tuscany (Italy). Photochemistry and Photobiology, 87: 925-934. https://doi.org/10.1111/j.1751-1097.2011.00934.x

Author Response

Review 1: The article "Sunburns and Sun Protection Behaviors among Male Hispanic Outdoor Day Laborers" by Niu et al. consists of 14 pages. The abstract is well written and well summarizes the research. Research about outdoor day laborers and their patterns of sun-exposure behavior is of great importance but it is nothing new. Nevertheless, the interesting point of the study is combining the knowledge about sun protection with using sun protection in practice.

Response: Thank you for the positive comments on our manuscript. We now emphasize the unique contribution of this work (page 11).

The study examines different types of occupations, but do not consider, that different types of occupations mean different poses towards the sun. It also does not give the information about erythemal doses, that workers gained during their exposure. At least the information about maximum UV Index in the summer in the place of working should be included in the text. The paper should be reconsidered after the minor revision.

Response: Thank you for this comment. We agree that the different poses toward the sun could impact sun exposure differently. We have included this in our limitation and future directions (page 12).

Detailed comments:

2.1, page 3 - what was the period of the study? Which years were examined? What was the maximum UV Index?

Response: This study was conducted from October 2014 to April 2015. The average UV index in New Jersey the summer before data collection was "very high" at approximately 9.  We have added this information on page 3.

Line 120, page 3: Why the authors did not use the Fitzpatrick scale?

Response: Thanks. This measure we used had been proved as solid in a previous work focusing on the Hispanics in JAMA dermatology (Page 3).

Line 241, page 8: Authors explained the shortcut AOR, but not others (CI, B, S.E., Exp(B)). These shortcuts may be obvious to the authors, but not to the readers, who are not familiar with this type of regression.

Response: Thank you for this comment. We have explained the terms in the paper and added those in the table (page 9-10).

Line 338, page 12: It is not novel to investigate different types of occupations. See the references (and also the list of references in these papers):

Schmalwieser, A.W.; Casale, G.R.; Colosimo, A.; Schmalwieser, S.S.; Siani, A.M. Review on Occupational Personal Solar UV Exposure Measurements. Atmosphere 202112, 142. https://doi.org/10.3390/atmos12020142 

Siani, A.M., Casale, G.R., Sisto, R., Colosimo, A., Lang, C.A. and Kimlin, M.G. (2011), Occupational Exposures to Solar Ultraviolet Radiation of Vineyard Workers in Tuscany (Italy). Photochemistry and Photobiology, 87: 925-934. https://doi.org/10.1111/j.1751-1097.2011.00934.x

Response: Thank you. We have changed the phrase and cited these papers (page 12).

Reviewer 2 Report

The research performed in this manuscript is important in the light of skin cancer prevention, focusing on increasing knowledge about awareness of sun protection for/among a vulnerable, understudied group. The findings present interesting directions, e.g. the specific heritage of participants and sunburn risk as well as experience of heat illness. Also, focusing more on hat-wearing in interventions can be worthwhile as shown in this study. The manuscript is well written and of high quality and provides directions for interventions. The previous studies conducted by the authors regarding this target group shows a great relevance as well.

Some minor spelling checks, more subsantiation in some parts and shortening of sentences are recommended.

Author Response

Review 2: The research performed in this manuscript is important in the light of skin cancer prevention, focusing on increasing knowledge about awareness of sun protection for/among a vulnerable, understudied group. The findings present interesting directions, e.g. the specific heritage of participants and sunburn risk as well as experience of heat illness. Also, focusing more on hat-wearing in interventions can be worthwhile as shown in this study. The manuscript is well written and of high quality and provides directions for interventions. The previous studies conducted by the authors regarding this target group shows a great relevance as well.

Some minor spelling checks, more subsantiation in some parts and shortening of sentences are recommended.

Response: Thank you for the positive comments on our manuscript. We have worked on the minor errors and long sentences.
